# Description of Trends over the Week in Alcohol-Related Ambulance Attendance Data

**DOI:** 10.3390/ijerph20085583

**Published:** 2023-04-19

**Authors:** Kerri Coomber, Peter G. Miller, Jessica J. Killian, Rowan P. Ogeil, Naomi Beard, Dan I. Lubman, Ryan Baldwin, Karen Smith, Debbie Scott

**Affiliations:** 1School of Psychology, Deakin University, Geelong, VIC 3220, Australia; 2Turning Point, Eastern Health, Richmond, VIC 3121, Australia; 3Monash Addiction Research Centre, Eastern Health Clinical School, Monash University, Box Hill, VIC 3199, Australia; 4Centre for Research and Evaluation, Ambulance Victoria, Blackburn North, VIC 3130, Australia; 5School of Public Health and Preventive Medicine, Melbourne, VIC 3004, Australia; 6School of Primary Health Care, Monash University, Frankston, VIC 3199, Australia

**Keywords:** alcohol, ambulance attendances, temporal trends, Australia

## Abstract

Alcohol harms are often determined using a proxy measure based on temporal patterns during the week when harms are most likely to occur. This study utilised coded Australian ambulance data from the Victorian arm of the National Ambulance Surveillance System (NASS) to investigate temporal patterns across the week for alcohol-related ambulance attendances in 2019. These patterns were examined by season, regionality, gender, and age group. We found clear temporal peaks: from Friday 6:00 p.m. to Saturday 3:59 a.m. for both alcohol-involved and alcohol-intoxication-related attendance, from Saturday 6:00 p.m. to Sunday 4:59 a.m. for alcohol-involved attendances, and from Saturday 5:00 p.m. to Sunday 4:49 a.m. for alcohol-intoxication-related attendances. However, these temporal trends varied across age groups. Additionally, hours during Thursday and Sunday evenings also demonstrated peaks in attendances. There were no substantive differences between genders. Younger age groups (18–24 and 25–29 years) had a peak of alcohol-related attendances from 7:00 p.m. to 7:59 a.m. on Friday and Saturday nights, whereas the peak in attendances for 50–59 and 60+ years was from 5:00 p.m. to 2:59 a.m. on Friday and Saturday nights. These findings further the understanding of the impacts of alcohol during different times throughout the week, which can guide targeted policy responses regarding alcohol use and health service capacity planning.

## 1. Introduction

Globally, alcohol use is the leading behavioural risk factor for premature mortality and morbidity among people aged from 15 to 49 years, with 10% of all deaths between 1990 and 2016 in this age group attributable to alcohol [1,2]. Harms can either be acute (e.g., assault, road crashes, or suicide) or chronic (e.g., alcohol dependence, liver cirrhosis, or cancer). Harmful alcohol use (typically defined as patterns of alcohol use leading to detrimental physical or mental health outcomes) places preventable burden on the health and justice systems, for both the individual drinker and through harms to others [3], with particular times of the week associated with greater risk of acute harms [4].

Due to frequent unreliable recording of alcohol use or intoxication in health or justice data, the use of high-alcohol hours (HAH; hours during which harms from alcohol use peak) as a proxy for acute alcohol-related harms is often used [5,6,7,8]. HAH denotes when the majority of alcohol-related harms occurs and frontline services demand is at its highest [9,10]. The unreliable recording of alcohol use or intoxication within health data has resulted in misleading trends in harms, whereas data using HAH (which are not impacted by variation in staff recording practices) provide more reliable trends [11]. Therefore, the application of HAH is useful where actual alcohol use has not been recorded or is recorded inconsistently. When determining the hours that encompass HAH, research examining police recorded assaults in Australia indicated that 8:00 p.m.–6:00 a.m. [12,13] and 9:00 p.m.–6:00 a.m. [10] on Fridays and Saturdays reflected peaks in alcohol-related assaults (54% to 65%). Emergency department data demonstrated peaks in presentations reporting alcohol consumption in the six hours prior to an injury occurring between 12:00 a.m. and 5:00 a.m. on Fridays and Sundays [9]. For alcohol-related road traffic crashes, the pattern was more varied, although weekend late-night hours still typically presented greater risk [12,14].

While one paper indicated that ambulance attendances involving drug and alcohol use and aggression are more likely during HAH [15], this temporal pattern has yet to be examined in detail beyond cases involving violence and aggression. Ambulance data provide important insights into acute alcohol-related harms that are not routinely captured within burden of disease frameworks [16], and capture cases that are not attended to by police or are serious enough for transportation to an emergency department [17,18]. For instance, one UK study found that between 66% and 90% of violence-related ambulance attendances were not recorded in police data [19]. An additional strength of ambulance data is that services in Australia typically have wide coverage, of both population and geography, which may have been lacking in previous studies utilising HAH (i.e., data being confined to one city; [13]).

Given the lack of insight into patterns of ambulance attendances across the week, the current study expands upon the existing literature using emergency department and justice data, to explore the temporal trends of alcohol-related ambulance attendances within Victoria, Australia. As the data used in the current study to calculate and interpret these temporal trends do not contain attendances for non-alcohol-related presentations, a formal validation of HAH cannot be conducted. Rather, the current study provides a description of the patterns in alcohol-related ambulance attendances that could reflect HAH and could be considered proxy measures of HAH. Given alcohol consumption and related harms within regional areas are higher than those within metropolitan areas [20,21], temporal patterns in attendance data by regionality is examined in this study. We also examine temporal patterns by gender and age, as a higher proportion of males in Australia consume alcohol at risky levels and experience more harms compared with females, and young adults (18–25 years) are the most likely age group to engage in risky drinking and experience harm [4]. Patterns by season will also be explored. This paper will, therefore, describe patterns of attendances over the course of a week within an adult population and provide in-depth detail of the trends in alcohol-related ambulance attendances for differing demographic categories.

## 2. Materials and Methods

### 2.1. Data

Ethics approval was obtained from Eastern Health Human Research Ethics Committee. The project used data from the Victorian arm of the National Ambulance Surveillance System, details of which are described elsewhere [16]. In Australia, ambulance services are state based. When a health emergency is identified, concerned individuals dial ‘000’ and are connected to a dispatch system. Specially trained operators responding to the call identify relevant clinical information, triage the call according to severity, and an ambulance is dispatched as required. Once on-scene, trained paramedics assess and stabilise the patient/s, and then determine if the patient needs further medical treatment. If further care is required, the patient will be transported to the nearest available emergency department for definitive medical care. During each attendance, paramedics carefully document relevant clinical and ‘on-scene’ details in the electronic patient care record (ePCR). Ambulance Victoria provide ePCRs for individual cases that are automatically filtered using a keyword search to identify attendances associated with alcohol or other drugs to the National Ambulance Surveillance System at Turning Point. These data extracts are provided on a monthly, ongoing basis. A team of research assistants scrutinise paramedic records to determine the role of alcohol and drugs, and capture these details using a purpose-built validated and systematic coding system. Data are routinely assessed by project staff for inter-coder reliability and adherence to coding protocols through an audit process [16]. Results of a recent audit have been described in more detail by Lubman et al. [16], however, of the 221,662 alcohol and other drug variables re-coded during this audit, only 470 differences were identified. The coding system captures over 140 variables, which includes date and time details, information about the patient (including physical condition and substances used), and the attendance scene. Data for all ages are received and coded, however, the current paper utilises data for those aged 18 and over only.

### 2.2. Alcohol Involvement

Ambulance attendances were coded as either alcohol-involved or alcohol-intoxication on the basis of whether it is reasonable to attribute immediate or recent (past 24-h) alcohol use to the attendance. Alcohol-involved attendances are where paramedics have recorded alcohol as being involved in the attendance, but the patient may or may not be intoxicated. These attendances include alcohol consumption ranging from a small amount (e.g., one glass of wine) to large amounts prior to the ambulance arriving on the scene. Alcohol-involved attendances can include patients with a low level of alcohol, and prior research has indicated that aggression responses are heightened even at low blood alcohol levels [22,23]. Furthermore, decision-making and co-ordination can be easily affected by just one glass of alcohol, especially when the person is fatigued or has other underlying health comorbidities. Therefore, such attendances are clinically relevant.

Alcohol-intoxication attendances are a subset of alcohol-involved attendances where the paramedic notes that the patient was visibly intoxicated, supported by patient-reported alcohol consumption. Where there is ambiguity, coders default to ‘alcohol-involved’ and, therefore, ‘alcohol-intoxication’ cases may be under-represented. Alcohol-intoxication attendances often relate to drunkenness and alcohol poisoning, but can also include those where alcohol is not the primary reason for the attendance, such as alcohol-related injuries, acute disease, and other drugs.

Included cases may also involve the use of other substances, however, these have not been differentiated within the current study. This approach accurately captures the complexities around alcohol use, and the examination of specific substance use and its combinations is outside the scope of this paper.

### 2.3. Design and Analysis

The study used longitudinal, incident-level ambulance attendance data to describe temporal trends across contextual and demographic categories. Hourly data (for those 18 years and over) were obtained for 2019. This time period was chosen so as to remove the influence of the COVID-19 pandemic. Individual case data were aggregated by day of attendance. Data points with fewer than five cases were obfuscated for privacy reasons. These data were then examined for peaks in alcohol-involved and alcohol-intoxication attendances. Data for all hours of the week across the year were ranked (highest to lowest) according to the number of attendances, with the top 25th percentile retained. Data within this top 25th percentile were then examined for consistent patterns in attendances by hour and day of the week, guided by the hours outlined in the existing literature [9,10,12,13]. This approach was used for both alcohol-involved and alcohol-intoxication ambulance attendances. Individual case data were also aggregated for each of the following categories: Victoria-wide; season; regionality (metropolitan Melbourne local government areas versus regional local government areas); gender (male and female); and age group (18–24, 25–29, 30–39, 40–49, 50–59, and 60+ years). These descriptive analyses were conducted using Excel. It must be noted that the hours within the top 25th percentile for each data series was not included where there was no consistent pattern (i.e., only one hour during a night). A series of sensitivity analyses using data from 2017 and 2018 were also conducted.

## 3. Results

Victoria-wide attendances during 2019 increased during night-time hours (as defined in prior HAH literature), and peaked over Friday and Saturday nights (see Figure 1). Appendix A provide the trends over the hours of the week for alcohol-involved and alcohol-intoxication attendances by season, regionality, gender, and age group. A similar pattern was observed Victoria-wide for season and the three demographic categorisations. However, there were indications of the peak in attendances being earlier in the night for older (50+ years) age groups.

We explored the proportion of attendances during HAH as defined by the previous literature (8:00 p.m. and 6:00 a.m. on Friday and Saturday nights). We found that 26% of alcohol-involved and 28% of alcohol-intoxication attendances for Victoria were during these hours, despite HAH representing 12% of hours during the week. A similar proportion of attendances were found for each demographic group during these hours. However, the proportion of attendances during these hours for younger age groups were higher than 25%: for 18–24 years, alcohol-involved was at 44% and alcohol-intoxication was at 49%; for 25–29 years, alcohol-involved was at 35% and alcohol-intoxication at 39%. We found that the proportion of attendances for older age groups was somewhat lower during these hours: 21% for alcohol-involved and 22% for alcohol-intoxication in 40–49 years, 18% and 19% for 50–59 years, and 19% and 20% for 60+ years, respectively.

When examining the top 25th percentile of hourly attendance numbers, the temporal patterns Victoria-wide were similar for alcohol-involved (Friday 6:00 p.m. to Saturday 3:59 a.m.; Saturday 6:00 p.m. to Sunday 4:59 a.m.) and alcohol-intoxication (Friday 6:00 p.m. to Saturday 3:59 a.m.; Saturday 5:00 p.m. to Sunday 4:49 a.m.). The patterns within each season and demographic categories were also broadly similar to those seen Victoria-wide (Table 1 and Table 2), however, alcohol-intoxication attendances typically occurred later in the night for younger age groups (18–24 and 24–29 years) and earlier for older age groups (50–59 and 60+ years).

As expected, Friday and Saturday nights were over-represented in the data, however, other nights (Thursday and Sunday) while representing fewer attendances also demonstrated consistent attendance patterns. Although the number of attendances each hour for these additional nights were still within the top 25th percentile of alcohol-involved and alcohol-intoxicated attendances, given the overall lower attendance numbers, these could potentially be interpreted as moderate temporal peaks (see Appendix A). Together, these hours within the top 25th percentile of attendances capture 34% to 50% of alcohol-involved attendances and 39% to 56% of the alcohol-intoxication attendances.

### Sensitivity Analysis

To ensure robustness of the patterns reported, data for 2017 and 2018 were examined. The proportions of alcohol-related attendances during pre-defined HAH in 2017 and 2018 were found to be within one percentage point of that for 2019. Furthermore, the temporal patterns for both alcohol-involved and alcohol-intoxication attendances within 2017 and 2018 were largely the same as those of the 2019 data (differed by 1 h maximum). Where larger differences occurred, these were not consistent. For instance, there was indication that for male alcohol-involved attendances, Wednesdays 6:00 p.m.–11:59 p.m. during 2017 could possibly be considered as a moderate peak. However, this was not evident for the 2018 data or alcohol-intoxication data.

## 4. Discussion

We aimed to explore temporal patterns across the week of alcohol-related ambulance attendances in Victoria, Australia. Although our data broadly demonstrated similar peak times for alcohol-related harms to HAH, as defined in the literature, there was some variability in these temporal trends, particularly when investigating different age groups.

The trends found in this study provides some indication that the typical hours of alcohol-related harm vary by demographics, particularly within age groups (e.g., 7:00 p.m. to 7:59 a.m. on Friday and Saturday nights for those aged from 18 to 24 years). Additionally, consideration of other weeknights (Thursday and Sunday) as representing times during which there may be moderate alcohol-related attendance peaks could be warranted. However, temporal patterns by season, regionality, and gender did not markedly differ to that of previously defined HAH [10,12,13]. This stability in trends demonstrates that the continued use of an HAH proxy to measure alcohol-related harms could be a feasible approach to capture alcohol-related harms, where alcohol use is not reliably measured. However, the COVID-19 pandemic may also have had ongoing impacts on patterns of attendances, beyond a reduction in attendances during lockdown periods, e.g., [24]. Therefore, further exploration of the use of temporal trends is needed by comparing peaks in alcohol-related ambulance attendances to non-alcohol related attendances, and more formal testing of the presence of HAH would be useful to further refine a proxy measure of harm.

### Limitations

This study was dependent on details contained in the ePCR, which may have been influenced by the quality, consistency, and reliability of record-keeping. Furthermore, the current study utilised data from one jurisdiction in Australia, and the associated cultural and social environment related to alcohol use in Australia may possibly limit generalisability. The use of data from one year (2019) also potentially limits generalisability, however, the sensitivity analyses indicated relative robustness of the temporal trends that were identified. Additionally, we do not have a comparative group with data on non-substance related attendances, however, other studies have shown differences in temporal pattern based on underlying condition [25]. It may also be that these data are impacted by rare instances of reduced ambulance availability or patients using alternative transport options to get to an emergency department. However, as ambulance call-out data trends have been shown to match actual attendances [26] and data for the current study have been collected over a long period of time, it is anticipated that such impacts on the data would be minimal. Finally, we did not differentiate between event types, such as traffic crashes, assault, or injury; future research could examine temporal patterns by event type.

## 5. Conclusions

While not a direct examination of HAH, our study indicated that temporal peaks over the week in alcohol-related attendance data may not be static across demographic groups. Therefore, more research is needed to determine if adaptive HAH models would provide greater sensitivity for capturing harms. These findings also further the understanding of the impacts of alcohol during different times throughout the week, which can guide targeted policy responses regarding alcohol use and health service capacity planning.

## Figures and Tables

**Figure 1 ijerph-20-05583-f001:**
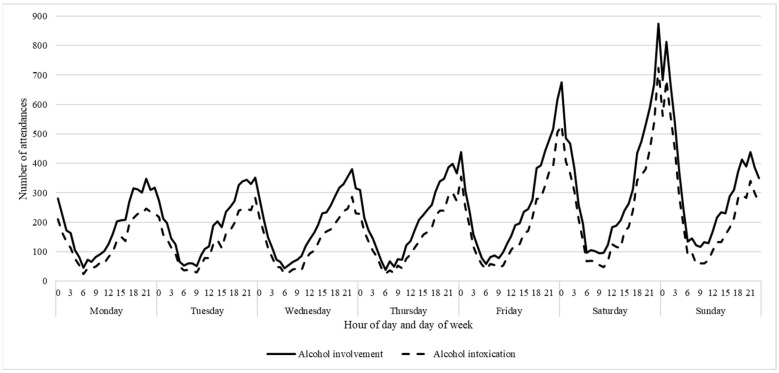
Alcohol-involved and alcohol-intoxication ambulance attendances across different times and days of the week in Victoria.

**Table 1 ijerph-20-05583-t001:** Indicative temporal peaks within the top 25th percentile of Victorian alcohol-involved ambulance attendance data, 2019.

**Season**	**Summer**		**Autumn**		**Winter**		**Spring**	
	Fri 6:00 p.m. to Sat 3:59 a.m.		Fri 6:00 p.m. to 3:59 a.m.		Fri 6:00 p.m. to Sun 3:59 a.m.		Fri 6:00 p.m. to Sat 3:59 a.m.	
	Sat 6:00 p.m. to Sun 4:59 a.m.		Sat 5:00 p.m. to Sun 4:59 a.m.		Sat 5:00 p.m. to Sun 4:59 a.m.		Sat 6:00 p.m. to Sun 4:59 a.m.	
**Region**	**Metropolitan**				**Regional**			
	Fri 6:00 p.m. to Sat 3:59 a.m.				Fri 6:00 p.m. to Sat 3:59 a.m.			
	Sat 5:00 p.m. to Sun 4:59 a.m.				Sat 6:00 p.m. to Sun 4:59 a.m.			
**Gender**	**Males**				**Females**			
	Fri 6:00 p.m. to Sat 3:59 a.m.				Fri 6:00 p.m. to Sat 4:59 a.m.			
	Sat 6:00 p.m. to Sun 4:59 a.m.				Sat 6:00 p.m. to Sun 4:59 a.m.			
**Age**	**18–24 years**	**25–29 years**		**30–39 years**	**40–49 years**	**50–59 years**		**60+ years**
	Fri 9:00 p.m. to Sat 5:59 a.m.	Fri 7:00 p.m. to Sat 5:59 a.m.		Fri 6:00 p.m. to Sat 3:59 a.m.	Fri 6:00 p.m. to Sat 2:59 a.m.	Fri 5:00 p.m. to Sat 0:59 a.m.		Fri 6:00 p.m. to Sat 0:59 a.m.
	Sat 7:00 p.m. to Sun 7:59 a.m.	Sat 6:00 p.m. to Sun 5:59 a.m.		Sat 6:00 p.m. to Sun 4:49 a.m.	Sat 5:00 p.m. to Sun 3:59 a.m.	Sat 6:00 p.m. to Sun 2:59 a.m.		Sat 5:00 p.m. to Sun 1:59 a.m.

Note: Fri = Friday; Sat = Saturday; Sun = Sunday.

**Table 2 ijerph-20-05583-t002:** Indicative temporal peaks within the top 25th percentile of Victorian alcohol-intoxication ambulance attendance data, 2019.

**Season**	**Summer**		**Autumn**		**Winter**		**Spring**	
	Fri 6:00 p.m. to 2:59 a.m.		Fri 6:00 p.m. to 3:59 a.m.		Fri 6:00 p.m. to Sun 3:59 a.m.		Fri 5:00 p.m. to Sat 4:59 a.m.	
	Sat 6:00 p.m. to Sun 4:59 a.m.		Sat 5:00 p.m. to Sun 4:59 a.m.		Sat 5:00 p.m. to Sun 4:59 a.m.		Sat 6:00 p.m. to Sun 4:59 a.m.	
**Region**	**Metropolitan**				**Regional**			
	Fri 6:00 p.m. to Sat 3:59 a.m.				Fri 6:00 p.m. to Sat 3:59 a.m.			
	Sat 5:00 p.m. to Sun 4:49 a.m.				Sat 6:00 p.m. to Sun 4:59 a.m.			
**Gender**	**Males**				**Females**			
	Fri 6:00 p.m. to Sat 3:59 a.m.				Fri 6:00 p.m. to Sat 3:59 a.m.			
	Sat 5:00 p.m. to Sun 4:49 a.m.				Sat 6:00 p.m. to Sun 4:59 a.m.			
**Age**	**18–24 years**	**25–29 years**		**30–39 years**	**40–49 years**	**50–59 years**		**60+ years**
	Fri 9:00 p.m. to Sat 5:59 a.m.	Fri 7:00 p.m. to Sat 4:49 a.m.		Fri 6:00 p.m. to Sat 3:59 a.m.	Fri 6:00 p.m. to Sat 2:59 a.m.	Fri 5:00 p.m. to Sat 0:59 a.m.		Fri 6:00 p.m. to Sat 0:59 a.m.
	Sat 7:00 p.m. to Sun 6:59 a.m.	Sat 5:00 p.m. to Sun 5:59 a.m.		Sat 6:00 p.m. to Sun 5:59 a.m.	Sat 5:00 p.m. to Sun 3:59 a.m.	Sat 6:00 p.m. to Sun 2:59 a.m.		Sat 5:00 p.m. to 11:59 p.m.

Note: Fri = Friday; Sat = Saturday; Sun = Sunday.

## Data Availability

Data sharing is not available due ethics approval restrictions.

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
