# Peer review of "Description of Trends over the Week in Alcohol-Related Ambulance Attendance Data"

_ijerph, 2023, doi:10.3390/ijerph20085583_

Round 1

Reviewer 1 Report

Dear authors,

Thank you for writing this article. The introduction is well-written.  A good methodology and analysis were performed. And the discussion is acceptable. Please see my comments below:

  • The novelty of the work would be better stated in details. The strengths of performing this study in Australia Or there is a study by Mason et al. (https://pubmed.ncbi.nlm.nih.gov/35231136/) reporting that the pattern of ambulance attendances for alcohol or other drug use changed during the COVID-19 lockdown period. COVID may still affect ambulance attendances for alcohol use. It is suggested that this study be cited both in the introduction and the discussion section.
  • State the exact design of the study in the section on methods.
  • What software was used to conduct the analysis?State the exact name of the analysis in the methods.
  • How was the use of other substances, such as drugs, monitored and considered in your analysis?
  • What if intoxication and alcohol use were reported for populations under 18? Did you not include them? So it must be stated as an inclusion criteria.
  • Please also indicate that ethnicity or cultural variations may also affect the results. These must be considered in the limitations. 

The main objective of the study was to explore temporal patterns across the week of alcohol-related ambulance 185 attendances in Victoria, Australia. Data broadly demonstrated similar peak times for alcohol-related harms as defined in prior literature, however they reported some variably in these temporal trends, particularly when investigating different age groups. (please see my comment No. 1 to the authors).

Overall, the novelty of the work is rare, except for some variably in the temporal trends, when investigating different age groups. (These are stated in the previous comment and my comment No. 1 to the author. )

As I mentioned in my comments, the manuscript is well-written but lacks much novelty.

Reviewer 2 Report

The manuscript titled "Description of trends over the week in alcohol-related ambulance attendance data" authored by Coomber et al. conducted a study to understand the impacts of alcohol consumption during different times throughout the week, which can guide targeted policy responses regarding alcohol use and health service capacity planning in Australia. While the study was conducted only in the Victoria province of Australia, the presented data would serve as a reference for country-wide studies. The study design is appropriate and methodology described is reproducible, results are discussed clearly. Conclusions drawn are adequate. Overall, the manuscript is written very well and I really enjoyed reading it. 

The authors have done a good job. This reviewer does not have any suggestions for the authors. 

The study investigated the impacts of alcohol consumption during different times throughout the week, which can guide targeted policy responses regarding alcohol use.   In my opinion, the study is interesting and would be useful for the health service capacity review and planning in Australia.   While there are other similar studies published in recent years in various countries, this study was conducted in the Victoria province of Australia and would serve as the reference data for a broader nationwide study. While other available studies might be localized up to the city level, this study using ambulance attendances provides a broader coverage.    This study appears to have provided a broader perspective on the alcohol-related ambulance attendances because the strength of ambulance data is services in Australia typically have wide coverage, for both population and geography, which may have been lacking in previous studies.    I find the manuscript is well written and organized. The text is clear to read and comprehend.    The conclusions drawn are consistent with the evidence and the arguments presented.   The conclusions address the study questions.  

Reviewer 3 Report

I am grateful for the opportunity to review your research article. This is an important piece of work and a good contribution to the alcohol research field. However, I would like to share some suggestions to further improve the manuscript and the presentation of the research conducted. Most importantly, a discussion of the research findings is largely missing. There is no discussion of observed patterns in HAH, nor of unobserved differences between seasons, regions and gender. Thus, the interesting results lose significant value given that they are placed without context and scientific standards are not met. My detailed suggestions are written below in the order of their appearance in the text.

I wish the authors good luck with their submission!

Introduction:

·      The alcohol burden statement in the very first sentence needs some more context (i.e., the year and country/region where it applies). Globally, for the population aged 15-49 years, high systolic blood pressure was *the* leading cause of death in 2019 (see GBD compare: https://vizhub.healthdata.org/gbd-compare/). Alcohol was the number two in the list of risk factors (but the first one within the category ‘behavioural’ risk factor).

·      I think the term “unnecessary burden” is a bit problematic as it is evaluative (though I am a non-native speaker). I would rather state that the burden is avoidable or preventable. You may also want to refer to harms people can experience from other’s alcohol use (see also the new release of “Alcohol: No Ordinary Commodity” by Babor et al. 2022).

·      In think the article would benefit from some more explanations why HAH can be considered a better(?) proxy for alcohol-related harm than other measures, such as alcohol intoxication data from EDs. This aspect is not clear to me and currently not well explained. From what I understand, however, HAH is a great measure to combine information on (acute) alcohol-related harm and peak hours of consumption to inform, e.g., prevention and intervention. Later, on page 2, you stress the aspect of HAH and *acute* alcohol-related harms. Overall, I think the introduction would benefit from some more explanations and some restructuring. It is currently a bit difficult to follow the line of argumentation.

Methods:

·      Data section: Please provide at least some information on the data source employed, including whether data was used at an individual level or aggregated. Moreover, please name the “validated and systematic coding system” and provide a reference, if available. Please also give more details on reliability and adherence to coding protocols – what are the results of these checks and is there any criteria used to ensure high quality?

·      Analysis: I guess all the additional information used in the analysis were available in through the attendance records as well? This is not explicitly stated.

Results:

·      Figure 1: Day-specific data was aggregated across seasons – is this correct? This is not entirely clear to me.

·      The results would substantially benefit when using statistical inference to evaluate significant changed across groups. Moreover, if possible, it would be more than useful to calculate rates to interpret geographical patterns, as well as patterns across sociodemographic groups.

·      Where time periods during night times determined ad-hoc? If so, it would be helpful to explicitly indicate these periods in the method section.

·      Seasonal and regional patterns are not indicated in the text. It is mandatory to at least refer to the fact that no differences where observed.

·      The sensitivity analysis should have already been introduced in the method section.

Discussion:

·      Sorry, but I am not convinced by the second paragraph of the first section in the discussion. You argue for tailored (policy?) responses based on the observed age pattern but it is neither clear to me what you mean by this nor you have yet discussed the validity of your estimates.

·      I further miss the discussion of your results. This includes not only why you have observed these age patterns but also why you may have not observed seasonal, geographical or gender-based patterns.

·      Finally, you conclude that the application of HAH as a proxy can be useful. However, you do not employ a method-focused analysis that allow to draw any conclusions regarding the use of HAH as a methodology. In my view, your aims and conclusion does not align, neither are your findings discussed or somewhere related to it.
